# Effect of Metal Dopant on the Performance of Ni@CeMeO$_2$ Embedded Catalysts (Me = Gd, Sm and Zr) for Dry Reforming of Methane

**André L. A. Marinho** [1,2,3], **Raimundo C. Rabelo-Neto** [2], **Florence Epron** [3], **Fabio S. Toniolo** [1], **Fabio B. Noronha** [2,4,*] **and Nicolas Bion** [3,*]

1    Chemical Engineering Program of COPPE/UFRJ, Federal University of Rio de Janeiro, P.O. Box 68502, Rio de Janeiro 21941-972, RJ, Brazil
2    National Institute of Technology, Catalysis Division, Rio de Janeiro 20081-312, RJ, Brazil
3    Institut de Chimie des Milieux et Matériaux de Poitiers (IC2MP), CNRS, University of Poitiers, 86073 Poitiers, France
4    University of Lille, CNRS, Centrale Lille, ENSCL, Univ. Artois, UMR 8181—UCCS—Unité de Catalyse et Chimie du Solide, 59000 Lille, France
\*    Correspondence: fabio.bellot@int.gov.br (F.B.N.); nicolas.bion@univ-poitiers.fr (N.B.)

**Abstract:** Biogas upgrading by a catalytic process has been studied in order to obtain syngas using renewable source of methane. This work evaluates the influence of metal dopant (Gd, Sm, and Zr) on the CeO$_2$ structure for the dry reforming of methane over Ni nanoparticle embedded catalysts. The doping with Zr improved the thermal stability of the catalyst, leading to the formation of small Ni nanoparticles, while Ni metal sintering was observed for Ni@CeO$_2$, Ni@CeGdO$_2$, and Ni@SmO$_2$, according to in situ XRD under reduction conditions. The ceria reducibility was affected by the dopant nature, for which the addition of Zr caused distortions in the ceria lattice, promoting the diffusion of oxygen bulk to surface. The doping with Gd and Sm created oxygen vacancies by charge compensation, and the saturation of oxygen vacancies in the fresh samples decreased the degree of Ce reduction, according to TPR results. The larger Ni particles and poor redox behavior for Ni@CeGdO$_2$ and Ni@CeSmO$_2$ were responsible for the high carbon formation on these catalysts during the DRM reaction. The Ni@CeZrO$_2$ catalyst did not present coke formation because of smaller Ni crystallite size and higher ceria reducibility. Therefore, the control of Ni particle size and the high oxygen mobility in the Ni@CeZrO$_2$ catalyst inhibits carbon deposition and enhances the mechanism of carbon removal, promoting the catalyst stability.

**Keywords:** Ni-embedded; ceria; ceria-zirconia; Gd-doped ceria; Sm-doped ceria; oxygen storage capacity; oxygen isotopic exchange; methane dry reforming; biogas

## 1. Introduction

Energy from renewable sources has gained much more attention in recent years due to the environmental impacts caused by greenhouse gases emission from fossil fuel [1,2]. Among the renewable sources, biogas plays a key role in the development of technologies to replace CH$_4$ dependency from the natural gas. Biogas is produced from the anaerobic digestion of biomass and it is a renewable source of methane and carbon dioxide [3]. Nowadays, biogas is mainly upgraded to biomethane and incorporated into natural gas networks [2]. Alternatively, biogas may be used for the production of syngas and hydrogen through the dry reforming of methane (DRM) [4,5]. This is an interesting route due to the presence of CH$_4$ and CO$_2$ in the raw biogas, decreasing the number of pretreatment steps before the reformer. Furthermore, CH$_4$ and CO$_2$ are greenhouse gases turning the DRM process environmentally friendly.

DRM is an endothermic reaction that requires high temperature (>800 °C), which favors catalyst deactivation by metal sintering and carbon deposition. Therefore, the design of a stable catalyst is the main challenge for the DRM.

In order to improve the resistance to metal sintering, many strategies were applied to improve the Ni dispersion and different methods of catalyst preparation have been investigated, such as evaporation-induced self-assembly, core-shell, and sol-gel, to limit the Ni crystallite growth [6–13]. Ma et al. recently demonstrated that the addition of La increased the Ni dispersion over $Al_2O_3$ support, avoiding the formation of $NiAl_2O_4$ species [13]. However, the TGA analysis already shows the formation of carbon after 10 h of reaction. Zhang et al. [12] studied the performance of a $Ni@SiO_2$ core-shell catalyst for DRM reaction. They observed the resistance to Ni sintering at 850 °C for 24.5 h on stream. However, $N_2$ physisorption measurements revealed that the porous structure of silica shell collapsed during reduction and reforming, which was attributed to the instability of silica shell under the high reaction temperature, especially in the presence of produced steam at high temperatures. This has also been reported in other studies using core–shell catalysts [14,15]. The formation of carbon filaments inside the nanocavities was inhibited but amorphous and graphitic carbon were still formed. Charisiou et al. demonstrated the effect of temperature on the type of carbon species formed during DRM reaction [16]. Performing the reaction above 750 °C, the carbon formed presents high graphitization degree and it is more easily oxidized by the O species over the $Ni/CeO_2$-$ZrO_2$ catalyst surface.

Therefore, control of metal sintering alone is not sufficient to prevent the formation of carbon deposits on Ni-based catalysts with unreducible supports. Furthermore, the synthesis methods for core-shell materials reported in the literature are still limited, present difficulty to control core size and shell thickness, and lead to low yield and high-price reactants [17]. These works reveal the importance of developing more simple methods to prepare embedded materials.

Recently, Ni nanoparticles embedded in $CeO_2$ ($Ni@CeO_2$) were synthesized by the sol-gel method and compared with a $Ni/CeO_2$ prepared by support impregnation [6]. The performance of the catalysts was investigated in the DRM reaction. Ni embedded in $CeO_2$ improved the resistance to sintering along the reduction at 800 °C and reduced the carbon formation rates during DRM reaction at 800 °C. These results were attributed to the higher metal–support interaction and larger amount of oxygen vacancies.

Ceria ($CeO_2$) has been extensively used as a support of catalysts for DRM reaction due to its redox properties [18–21]. Ceria has a very high oxygen exchange capacity since this oxide may reversibly change its oxidation states between $Ce^{4+}$ and $Ce^{3+}$, resulting in the generation of oxygen vacancies [22]. Therefore, at DRM conditions, the carbon deposits can be gasified by lattice oxygen from $CeO_2$ and the $CO_2$ dissociates on oxygen vacancies, promoting the carbon removal mechanism [23,24]. The oxygen storage capacity (OSC) can be influenced by ceria surface area and particle size and by the addition of dopants into $CeO_2$ structure [25–28].

The addition of dopants promotes the OSC of ceria-based materials [20,28–31]. The most common dopant found in the literature is zirconia. The substitution of $Zr^{4+}$ for $Ce^{4+}$ generates oxygen vacancies due to the distortion of ceria lattice [6,32–34]. Zirconia also promotes thermal stability of ceria, avoiding the sintering at high temperature [6,35]. Doping of ceria with lower valence rare-earth elements, such as $Gd^{3+}$ and $Sm^{3+}$, results in the formation of oxygen vacancy by charge compensation [28].

Luisetto et al. [20] studied the effect of Zr and Sm doped-$CeO_2$ over $Ni/CeO_2$ catalyst in the DRM reaction. They observed by Raman spectroscopy that the number of defects created by doping ceria with Sm was higher than that for Zr doping. Ni particle size was extremely affected by metal dopant, for which $Ni/CeZrO_2$ presented the smallest value (11 nm) against 37 nm for $Ni/CeSmO_2$, indicating the low thermal stability of Sm-doped ceria materials.

The aim of the present work is to investigate, for catalysts constituted of Ni-embedded in ceria support, the effect of different metal dopants ($Gd^{3+}$, $Sm^{3+}$ and $Zr^{4+}$) for limiting the carbon deposition. The effect of the type of dopant on the nature of oxygen vacancies (created by charge compensation or structural relaxation in the ceria structure) and on the rate of carbon formation during DRM reaction was evaluated. The reducibility and oxygen mobility of the catalysts were characterized by in-situ X-ray diffraction, temperature-programed reduction, and isothermal oxygen isotopic exchange experiments. The latter characterization technique, not classically used for investigating DRM catalysts, is well suited to study the impact of dopants in the diffusion of $O^{2-}$ from the ceria bulk to the surface where the carbon deposition occurs. The carbon deposits were measured by thermogravimetric analysis and scanning electron microscopy.

## 2. Results and Discussion

### 2.1. Catalyst Characterization

The chemical composition and surface area of the samples are reported in Table 1. The Ni content and Ce/dopant molar ratio are close to the nominal value (i.e., Ni wt.% = 10 and Ce/dopant molar ratio = 4.0). After reduction at 800 °C under pure $H_2$, all catalysts exhibited very low surface area due to the low thermal stability of ceria at this high reduction temperature [22]. The addition of Gd or Sm to ceria did not change the thermal stability of the catalysts, which showed surface area lower than 10 $m^2/g$. Silva et al. [31] also did not observe in their work any changes in the surface area for $Ni/CeGdO_2$ compared to $Ni/CeO_2$. The addition of Zr slightly increased the thermal stability of the catalyst, presenting the highest surface area among them. Several works in the literature [35–37] also reported the beneficial effect of Zr to enhance $CeO_2$ thermal stability by the formation of a $CeZrO_2$ solid solution.

**Table 1.** Chemical composition and surface area after reduction at 800 °C of the catalysts.

| Catalyst | Ni (wt%) | CeO$_2$ (wt%) | Dopant (wt%) | Ce/Dopant Molar Ratio | Surface Area of Reduced Sample (m$^2$/g) |
|---|---|---|---|---|---|
| Ni@CeO$_2$ | 9.7 | 87.2 | — | — | <10 |
| Ni@CeGdO$_2$ | 9.8 | 72.1 | 15.1 | 4.2 | <10 |
| Ni@CeSmO$_2$ | 9.7 | 72.4 | 17.9 | 4.1 | <10 |
| Ni@CeZrO$_2$ | 9.5 | 74.4 | 13.4 | 4.0 | 20 |

Figure 1a shows the diffractograms of the fresh samples at room temperature. The XRD patterns were converted to $CuK\alpha$ radiation ($\lambda$ = 1.5406 Å), for direct comparison with the literature. The line at $2\theta$ = 44.5° is due to the Khantal-based support used in the in situ XRD equipment. The diffractogram of $Ni@CeO_2$ catalyst exhibited the characteristic lines of $CeO_2$ with fluorite-like structure (JCPDS 34-0394) at $2\theta$ = 28.46°, 32.92°, 47.39°, and 56.33°. These lines are marked with dashed lines to facilitate the discussion about the shifts observed with the addition of different dopants. The insertion of dopants shifts the main $CeO_2$ line to lower ($Ni@CeGdO_2$ = 28.37°, $Ni@CeSmO_2$ = 28.33°) or higher ($Ni@CeZrO_2$ = 28.79°) $2\theta$ values (Figure 1b) as a consequence of expansion or contraction of the ceria lattice, respectively. The ceria lattice for the calcined samples was: $Ni@CeO_2$ (5.4237 Å); $Ni@CeGdO_2$ (5.4460 Å); $Ni@CeSmO_2$ (5.4565 Å); $Ni@CeZrO_2$ (5.3681 Å). The partial substitution of $Ce^{4+}$ (0.97 Å) by cations with larger atomic radii, such as $Gd^{3+}$ (1.16 Å) or $Sm^{3+}$ (1.08 Å), causes an expansion in the ceria lattice [38]. The $Zr^{4+}$ addition leads to a contraction of the ceria lattice since its atomic radius (0.84 Å) is smaller than that for $Ce^{4+}$ (0.97 Å). In our work, the XRD of fresh samples reveals that Gd, Sm, and Zr partially substitute $Ce^{4+}$ into the structure, leading to the formation of $CeGdO_2$, $CeSmO_2$, and $CeZrO_2$ solid solutions [32,39–41]. The lines characteristic of NiO were not detected in the diffractograms of the calcined samples, indicating the presence of highly dispersed particles, as observed in our previous work [6].

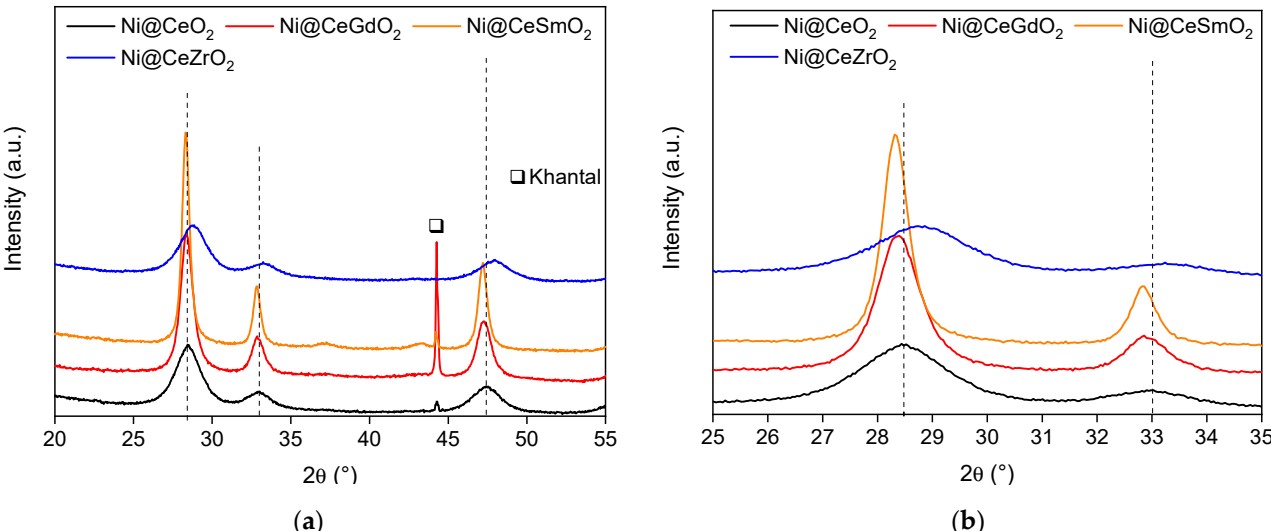

**Figure 1.** XRD patterns of fresh samples at room temperature: (**a**) 2θ = 20–55°; (**b**) 2θ = 25–35°.

In-situ diffractograms obtained during reduction of the samples are presented in Figure 2. For Ni@CeO$_2$, Ni@CeGdO$_2$, and Ni@CeSmO$_2$, the positions of the ceria lines remain unchanged during reduction up to 300 °C. Further heating up to 500 °C slightly shifts the diffraction lines to lower 2θ angles. Increasing the temperature up to 800 °C results in a significant shift of these lines. The shift of characteristic lines of ceria to lower 2θ angles during reduction of all samples could be due to: (i) the thermal expansion of the unit cell and/or; (ii) the reduction of Ce$^{4+}$ to Ce$^{3+}$, which has larger ionic radii (1.14 Å) [42]. Therefore, the shift on the ceria lines observed during the reduction process is a result of the balance between these two effects. On the other hand, for the Ni@CeZrO$_2$ catalyst, a shift in the ceria lines is observed when the sample is reduced at 300 °C, and this displacement increases when the sample is heated up to 800 °C. The formation of CeZrO$_2$ solid solution favors ceria reduction at lower temperature compared to Ce-doped with Gd or Sm.

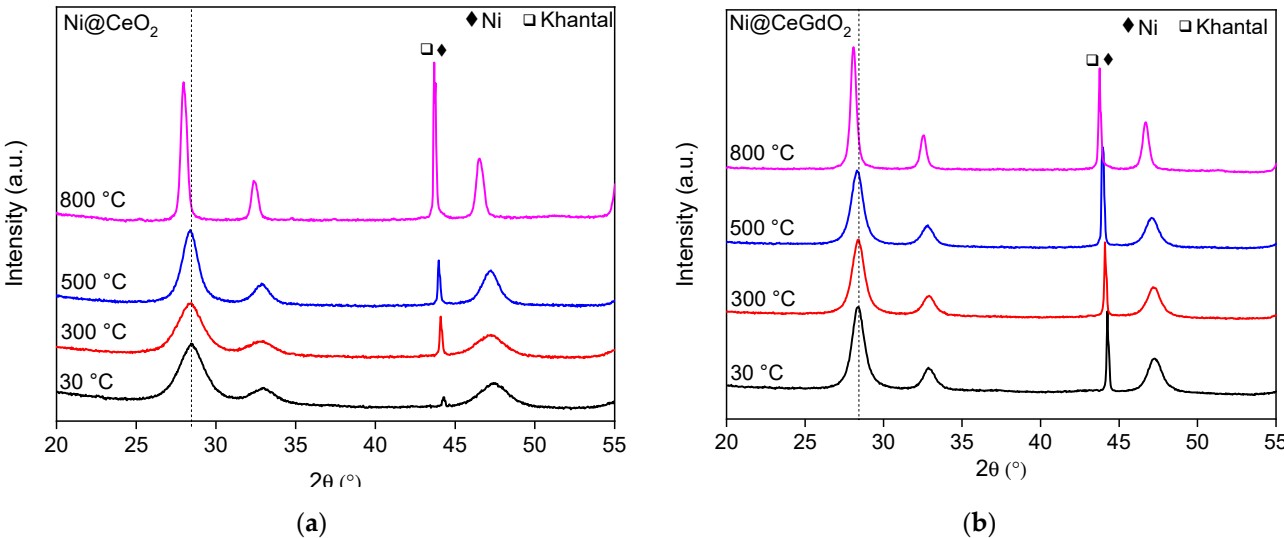

**Figure 2.** *Cont.*

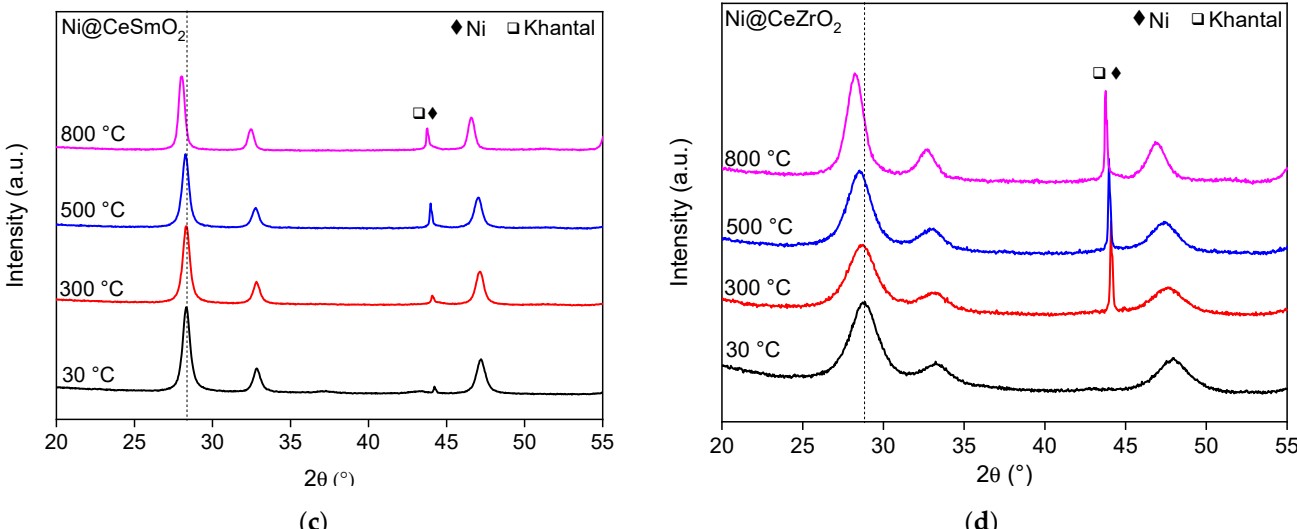

**Figure 2.** In-situ diffractograms obtained during reduction of (**a**) Ni@CeO$_2$, (**b**) Ni@CeGdO$_2$, (**c**) Ni@CeSmO$_2$ and (**d**) Ni@CeZrO$_2$.

In-situ XRD experiments under air of a NiO-CeO$_2$ physical mixture up to 800 °C (not shown) were carried out to evaluate the influence of the thermal expansion on the ceria lattice parameter. The thermal expansion was responsible for an increase in the CeO$_2$ lattice of 0.04 Å. Thus, a variation in this parameter higher than that value can be associated with the formation of Ce$^{3+}$ by ceria reduction.

Figure 3a shows the variation of the calculated ceria lattice parameter as a function of reduction temperature for all catalysts. Increasing the temperature up to 800 °C, it is observed a slight increase in the ceria lattice parameter for Ni@CeO$_2$, Ni@CeGdO$_2$ and Ni@CeSmO$_2$ catalysts. However, the increase is much more important when the CeZrO$_2$ solid solution is formed. The ceria lattice expansion follows the order: Ni@CeGdO$_2$ (0.0556 Å) < Ni@CeSmO$_2$ (0.0656 Å) < Ni@CeO$_2$ (0.0924 Å) < Ni@CeZrO$_2$ (0.1011 Å). Therefore, the values of ceria lattice expansion were higher than 0.04 Å, indicating the formation of Ce$^{3+}$ by ceria reduction for all catalysts. However, the reduction degree of ceria depends on the type of dopant. The XRD data suggest that the doping with Gd and Sm inhibits ceria reduction while the presence of Zr promotes the reduction of CeO$_2$.

Figure 3b shows the evolution of ceria crystallite size during reduction. The calcined samples doped with Gd and Sm exhibit the largest ceria crystallite size. Ni@CeO$_2$ and Ni@CeGdO$_2$ catalysts undergo a significant growth of the ceria crystallite size above 500 °C, but the sintering degree was more important for the undoped sample. For the samples doped with Sm and Zr, the ceria crystallite size only slightly increases during reduction at 800 °C. These results demonstrate that doping ceria increases the resistance to sintering during reduction at high temperature and this effect is more pronounced for Sm and Zr. Ni@CeZrO$_2$ exhibits the smallest ceria crystallite size after reduction at 800 °C. The high thermal stability of the Zr-doped ceria catalysts has been attributed to the CeZrO$_2$ solid solution formation [35,43,44].

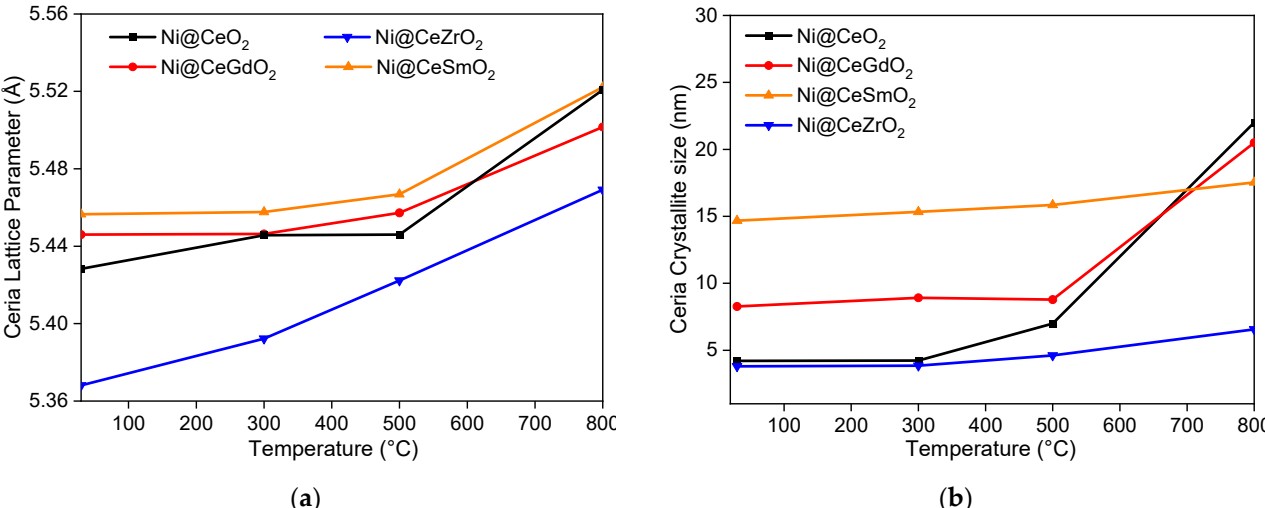

**Figure 3.** Variation of (**a**) ceria lattice parameter and (**b**) ceria crystallite size during reduction process.

Ni crystallite size was not calculated from the in situ diffractograms because the main characteristic line of metallic Ni is overlapped by Khantal line at $2\theta = 44.5°$, and the *(200)* plane of $Ni^0$ is characterized by a low intensity peak. Therefore, the Ni crystallite size was calculated by XRD after ex-situ reduction of the catalysts at 800 °C followed by passivation. The results are reported in Table 2. The Ni crystallite size is approximately the same for $Ni@CeO_2$, $Ni@CeGdO_2$, and $Ni@CeSmO_2$ (between 9.5 and 11.0 nm) while it is smaller for Zr-doped material (5.4 nm). The high resistance to $CeO_2$ sintering observed for Ce-doped with Zr catalysts avoids Ni sintering.

**Table 2.** Ni crystallite size calculated for the catalysts reduced at 800 °C followed by passivation and TOF values for the dry reforming of methane at 600 °C.

| Catalyst | Ni Crystallite Size (nm) | TOF ($s^{-1}$) |
|----------|--------------------------|----------------|
| $Ni@CeO_2$ | 10.7 | 3.6 |
| $Ni@CeGdO_2$ | 11.0 | 3.6 |
| $Ni@CeSmO_2$ | 9.5 | 3.7 |
| $Ni@CeZrO_2$ | 5.4 | 2.7 |

The Raman spectra of the calcined samples are shown in Figure 4. The sharp band at 460 $cm^{-1}$ is ascribed to the symmetrical stretching mode between the eight oxygen atoms bound to the cerium atom in the triple degenerate $F_{2g}$ mode. The addition of different dopants slightly shifted the position of the band to higher Raman shift, due the modification of Ce-O force constant, evidencing the solid solution formation as previously demonstrated by XRD. Besides this intense band, it is possible to observe for all catalysts the presence of additional bands at 549 and 634 $cm^{-1}$ assigned to defect-induced mode. The comparison with a $Ni/CeO_2$ catalyst prepared by classical impregnation was already reported in a previous study [6], and eliminated the possibility that these bands were due the NiO vibration modes.

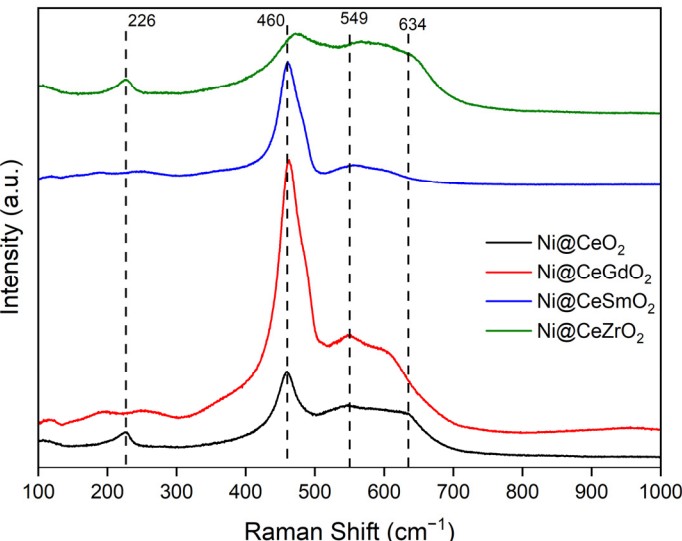

**Figure 4.** Raman spectra for the calcined samples.

Two different defects are reported in the literature: the Raman band at 549 cm$^{-1}$ is associated to defect space including $O^{2-}$ vacancy ($O_v$), while the band at 634 cm$^{-1}$ is associated to defects space without $O^{2-}$ vacancy [45,46]. Moreover, the Ni@CeO$_2$ and Ni@CeZrO$_2$ catalysts presented an additional band at 226 cm$^{-1}$ assigned to second-order transverse acoustic mode (2TA). Among the catalysts, it is possible to observe that the $F_{2g}$ band is much more intense for the catalysts doped with Gd and Sm. According to Loridant [47], the broadening observed in the Ni@CeO$_2$ and Ni@CeZrO$_2$ catalysts would be due to the presence of defects and to lower CeO$_2$ crystallite sizes observed for these catalysts. On the basis of the $O_v/F_{2g}$ intensity ratio as proposed in [46], one can conclude that the degree of defects associated to oxygen vacancies increases with the following order Ni@CeGdO$_2$ < Ni@CeSmO$_2$ < Ni@CeO$_2$ < Ni@CeZrO$_2$.

TPR analysis was performed to study the reducibility of each catalyst and the profiles are shown in Figure 5. The TPR profile of Ni@CeO$_2$ catalyst exhibits broad peaks at low temperature (below 600 °C) and above 800 °C. Ni@CeGdO$_2$ and Ni@CeSmO$_2$ catalysts show two peaks at 340 and 420 °C, but the peak at high temperature is no longer observed. Ni@CeZrO$_2$ catalyst shows peaks at 340 and 420 °C and a shoulder at 490 °C.

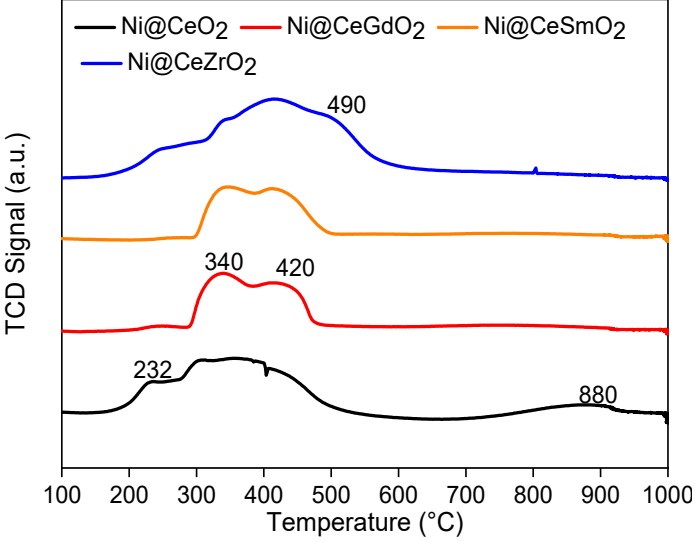

**Figure 5.** TPR profiles for the catalysts.

In situ XANES experiments at the Ni K-edge and Ce L$_{III}$ edge were carried out to investigate the reduction of Ni/CeO$_2$ catalysts [6,48]. The XANES analyses at the Ni K-edge reveal that NiO is completely reduced up to 750 °C. The reduction of Ce$^{4+}$ to Ce$^{3+}$ occurs in two different regions: the low temperature region (200–650 °C), which is associated with the reduction of surface ceria; and the reduction of bulk ceria at high temperature (above 650 °C). Therefore, the reduction of NiO and surface CeO$_2$ occurs simultaneously at the low temperature region. Once the metallic Ni particles are formed, hydrogen is dissociated on the Ni surface and spills over to ceria, promoting its reduction. In our work, the peaks below 600 °C can be attributed to the simultaneous reduction of NiO and surface CeO$_2$, while the peak at high temperature observed in the TPR profile of Ni@CeO$_2$ corresponds to the reduction of bulk CeO$_2$.

Table 3 reports the H$_2$ uptake and the reduction degree of ceria calculated from the TPR profiles, considering the complete reduction of NiO. The reduction degree of ceria varied in the following order: Ni@CeGdO$_2$ (7%) < Ni@CeSmO$_2$ (13%) < Ni@CeO$_2$ (20%) < Ni@CeZrO$_2$ (35%). This trend is exactly the same observed for the ceria lattice expansion calculated from the in situ diffractograms during the reduction. Therefore, these results confirm that Gd and Sm-doped ceria catalysts exhibit lower ceria reducibility than undoped ceria. Hennings and Reimert [49] also demonstrated by TPR measurements that the doping of CeO$_2$ with different Gd contents reduced the degree of reduction of ceria.

**Table 3.** H$_2$ uptake and CeO$_2$ reduction degree calculated from TPR.

| Catalyst | H$_2$ Uptake (μmol/g) | | Theoretical H$_2$ Consumption for Total Reduction (μmol/g) | | Ce$^{4+}$/Ce$^{3+}$ Reduction (%) |
| --- | --- | --- | --- | --- | --- |
| | Low Temperature | High Temperature | Ni$^{2+}$/Ni$^0$ | Ce$^{4+}$/Ce$^{3+}$ | |
| Ni@CeO$_2$ | 2022.7 | 643.3 | 1652.7 | 5066.4 | 20 |
| Ni@CeGdO$_2$ | 1963.0 | —— | 1669.7 | 4190.1 | 7 |
| Ni@CeSmO$_2$ | 2199.7 | —— | 1652.7 | 4207.5 | 13 |
| Ni@CeZrO$_2$ | 3131.4 | —— | 1618.5 | 4322.7 | 35 |

The addition of Zr strongly increases the Ce$^{4+}$ to Ce$^{3+}$ reduction from 20% (on Ni@CeO$_2$) to 35% (on Ni@CeZrO$_2$). The higher reducibility of Zr-doped catalysts is due to the formation of ceria-zirconia solid solution, which has higher oxygen mobility [34]. The CeZrO$_2$ solid solution is described in the literature as a highly reducible material and it has been extensively used as catalyst support for DRM reaction [50–52]. Chen et al. [53] studied the Ni/Ce$_{0.75}$Zr$_{0.25}$O$_2$ catalyst for DRM reaction and they also observed a high H$_2$ consumption for CeZrO$_2$ materials compared to pure CeO$_2$. The doping with Zr promotes the mobility of oxygen from bulk to surface, enhancing the ceria reducibility.

The experiments of isotopic oxygen exchange were carried out at 400 °C and the results showing the evolution of the number of exchanged atoms (N$_e$) for the Ce-doped materials are depicted in Figure 6. The Ni@CeO$_2$ catalyst presents a continuous increase on N$_e$ during the time exposed at $^{18}$O$_2$, reaching the value of $33 \times 10^{20}$ at·g$^{-1}$ after 90 min.

Comparing the different dopants, Gd and Sm rapidly reach the equilibrium value at 400 °C, around N$_e$ = $35 \times 10^{20}$ at·g$^{-1}$. The high rate of exchange obtained for these catalysts is due to the presence of high surface exchange activity. According to the literature [54,55], Gd and Sm segregate to the ceria surface and, therefore, the oxygen vacancies are more pronounced over the material surface, enriching the surface activity. In contrast, the Ni@CeZrO$_2$ catalyst presents a lower initial rate of exchange and slightly higher N$_e$ after 2 h ($37 \times 10^{20}$ at·g$^{-1}$) under $^{18}$O$_2$. The initial exchange rate, dependent on the concentration of surface sites able to activate the O$_2$ molecule, was not expected to be the lowest for this sample which exhibits the lowest crystallite size (highest specific surface area). This emphasizes the presence of preferential sites, e.g., oxygen vacancies for Gd and Sm-doped samples. On the contrary, the higher N$_e$ value observed for the Ni@CeZrO$_2$ catalyst confirms the role of Zr substitution in ceria to favor the bulk diffusion.

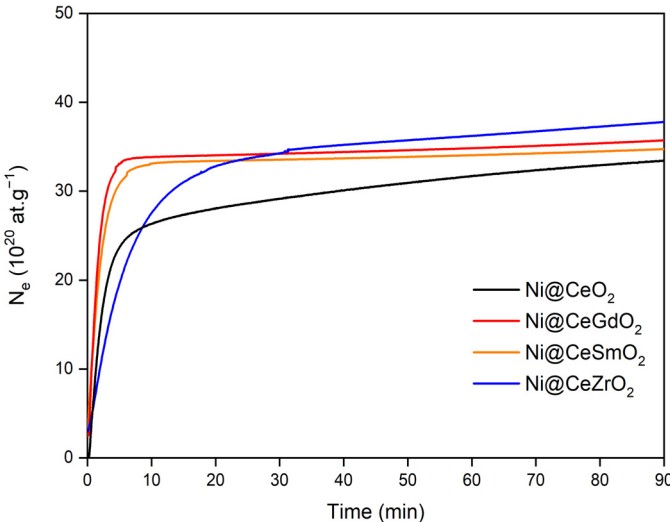

**Figure 6.** Evolution of the number of exchanged oxygen atoms during IOIE at 400 °C for Ni@CeO$_2$, Ni@CeGdO$_2$, Ni@CeSmO$_2$ and Ni@CeZrO$_2$.

Hennings and Reimert [49] observed that the total oxygen storage capacity (OSC) decreased with the increase in the Gd content in the ceria lattice. This result was attributed to the replacement of Ce$^{4+}$ ions by non-reducible Gd$^{3+}$ ions and the lower Ce$^{4+}$ conversion.

Our experiments point out that the doped-catalysts present similar N$_e$, but the oxygen exchange dynamic depends on the dopant metal. Doping with Gd and Sm favors the exchange only at surface level, while doping with Zr favors the participation of bulk oxygen species, which will increase the number of oxygen vacancies under reaction condition.

### 2.2. DRM Reaction

Figure 7 displays the conversions of CO$_2$ and CH$_4$ expected at the thermodynamic equilibrium as a function of temperature. This analysis was performed by using the Gibbs free energy minimization algorithm on Aspen HYSYS Software assuming no carbon formation. At 800 °C, the conversions of 95 and 90% are reached at the equilibrium for CO$_2$ and CH$_4$ respectively. The highest conversion for CO$_2$ is explained by the occurrence of the reverse water-gas shift reaction (RWGS: CO$_2$ + H$_2$ → CO + H$_2$O) which is an endothermic reaction thermodynamically favored at high temperature, converting the CO$_2$ in CO in the presence H$_2$. This side reaction leads to a decrease of the H$_2$/CO molar ratio in the produced syngas.

Table 2 lists the initial turnover frequency (TOF) values of the catalysts for the DRM at 600 °C calculated at low CH$_4$ conversion. All catalysts exhibit approximately the same TOF values (between 2.7 and 3.7 s$^{-1}$), which is reasonable since they have the same Ni dispersion. TOF values obtained in our work are in agreement with those reported by Wei and Iglesia [56] for Ni/MgO (4.0 s$^{-1}$—Ni crystallite size = 6.7 nm).

The stability test was carried out to evaluate the resistance of the catalysts to carbon deposition during the DRM. The operating conditions were adapted in order to avoid reaching the thermodynamic equilibrium, allowing a relevant comparison in terms of catalyst deactivation. CH$_4$ and CO$_2$ conversions and H$_2$/CO molar ratio are shown as a function of time on stream (TOS) in Figures 8 and 9, respectively. CH$_4$ and CO$_2$ conversions remain quite constant during 24 h of TOS for all catalysts. The CO$_2$ conversion is higher than CH$_4$ conversion for all catalysts while the H$_2$/CO molar ratio is lower than 1.0. These results suggest the occurrence of the reverse water-gas shift (RWGS) reaction, which is thermodynamically favored at the reaction conditions used. A similar result was reported by other authors [32,41,57]. The Gd and Sm-doped catalysts present higher initial H$_2$/CO molar ratio, followed by Ni@CeO$_2$ and Ni@CeZrO$_2$.

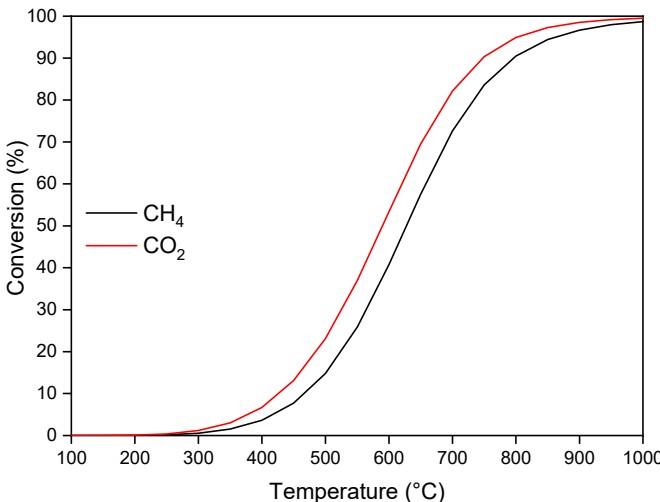

**Figure 7.** Reactant conversion and $H_2/CO$ molar ratio at the thermodynamic equilibrium of the DRM reaction assuming no carbon formation.

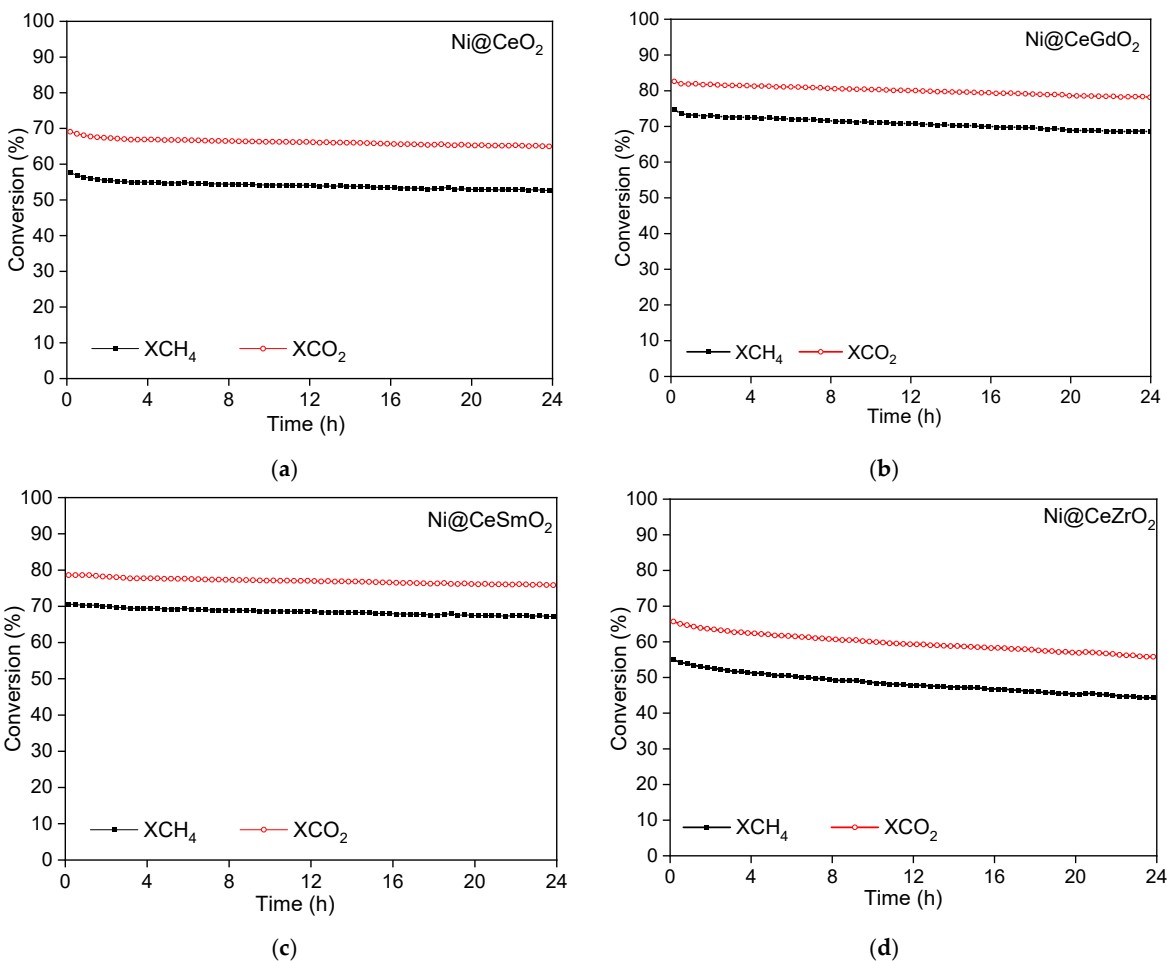

**Figure 8.** Conversion of $CH_4$ and $CO_2$ in the DRM at 800 °C as a function of TOS ($CH_4/CO_2$ molar ratio of 1.0; W/F = 0.2 $g_{cat} \cdot L^{-1} h^{-1}$) for (**a**) Ni@CeO$_2$, (**b**) Ni@CeGdO$_2$, (**c**) Ni@CeSmO$_2$ and (**d**) Ni@CeZrO$_2$.

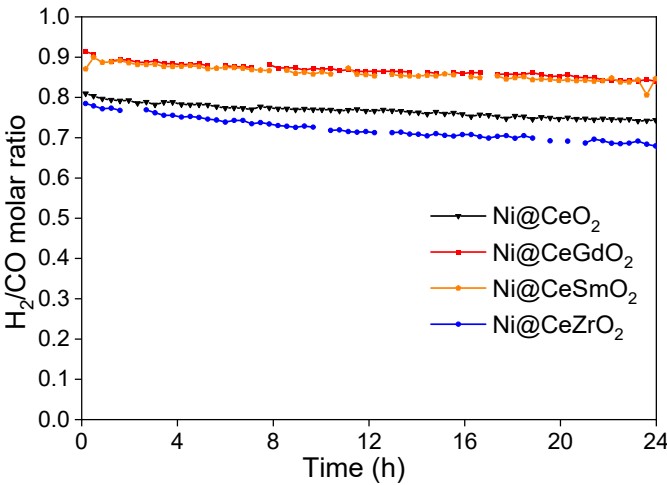

**Figure 9.** $H_2/CO$ molar ratio obtained in the the DRM at 800 °C as a function of TOS ($CH_4/CO_2$ molar ratio of 1.0; W/F = 0.2 $g_{cat} \cdot L^{-1} h^{-1}$).

## 2.3. Characterization of Spent Catalysts

After the stability test, the post-reaction samples were analyzed by TG and SEM to investigate the formation of carbon deposits. Figure 10 shows the DTG profile of each spent catalyst after DRM for 24 h of TOS. Ni@CeGdO$_2$ and Ni@CeSmO$_2$ exhibit an asymmetric peak at 604 °C with a shoulder at around 564–556 °C. The Ni@CeO$_2$ shows a broad and small peak at 503 °C and the Ni@CeZrO$_2$ catalyst does not show any peak in the TPO profile. According to the literature [6,48,58,59], the peak around 503 °C is ascribed to the oxidation of amorphous carbon, whereas the peak at 604 °C is due to the oxidation of carbon filaments with single or multiple walls. Table 4 presents the rate of carbon formation during the DRM for 24 h of TOS over the catalysts of our work and from the literature. Ni@CeGdO$_2$ has the highest carbon formation rate while no carbon formation is detected on Ni@CeZrO$_2$. High carbon formation was also observed over Cd and Sm-doped catalysts in the DRM reaction [60,61]. Taherian et al. [61] observed that the addition of Sm to SBA-15 induces the formation of carbon nanotubes over the support.

**Table 4.** Rate of carbon formation obtained by TGA in our work compared to the data obtained from literature.

| Catalyst | Reaction Conditions | Rate of Carbon Formation ($mgC \cdot gcat^{-1} \cdot h^{-1}$) | Reference |
|---|---|---|---|
| Ni@CeO$_2$ | 800 °C, CH$_4$:CO$_2$ = 1:1 | 1.6 | This work |
| Ni@CeGdO$_2$ | 800 °C, CH$_4$:CO$_2$ = 1:1 | 21.9 | This work |
| Ni@CeSmO$_2$ | 800 °C, CH$_4$:CO$_2$ = 1:1 | 9.8 | This work |
| Ni@CeZrO$_2$ | 800 °C, CH$_4$:CO$_2$ = 1:1 | 0.0 | This work |
| Ni/CeO$_2$ | 800 °C, CH$_4$:CO$_2$ = 1:1 | 9.7 | [6] |
| Ni/CeO$_2$ | 700 °C, CH$_4$:CO$_2$ = 1:1 | 0.4 | [34] |
| Ni/CeZrO$_2$ (75 wt% CeO$_2$) | 700 °C, CH$_4$:CO$_2$ = 1:1 | 3.5 | [34] |
| Ni/CeZrO$_2$ (44 wt% CeO$_2$) | 700 °C, CH$_4$:CO$_2$ = 1:1 | 1.7 | [34] |
| Ni/ CeZrO$_2$ (28 wt% CeO$_2$) | 700 °C, CH$_4$:CO$_2$ = 1:1 | 0.7 | [34] |
| NiCu/Ce$_{0.9}$Gd$_{0.1}$O$_2$ | 800 °C, CH$_4$:CO$_2$ = 1:1 | 12.2 | [60] |
| LaNiO$_3$ | 800 °C, CH$_4$:CO$_2$ = 1:1 | 27.0 | [58] |
| LaNiO$_3$/SiCeO$_2$ | 800 °C, CH$_4$:CO$_2$ = 1:1 | 0.3 | [58] |
| Ni/Gd-Y$_2$O$_3$ (1% Gd) | 700 °C, CH$_4$:CO$_2$ = 1:1 | 17.0 | [57] |
| Ni/Gd-Y$_2$O$_3$ (2% Gd) | 700 °C, CH$_4$:CO$_2$ = 1:1 | 14.6 | [57] |
| Ni/Gd-Y$_2$O$_3$ (3% Gd) | 700 °C, CH$_4$:CO$_2$ = 1:1 | 11.8 | [57] |

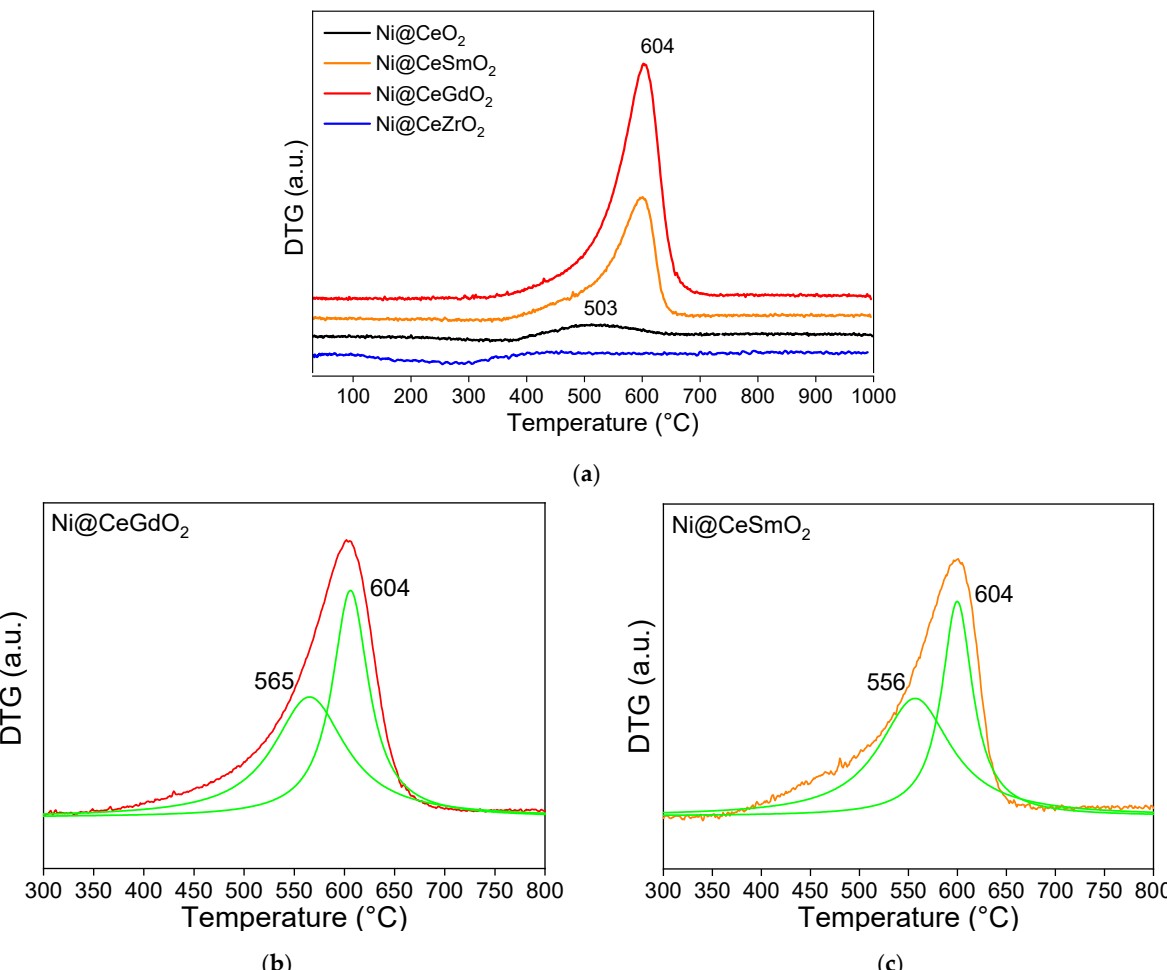

(a)

(b)

(c)

**Figure 10.** DTG profiles of (**a**) all used catalysts after DRM at 800 °C for 24 h of TOS with decomposition of the profiles for (**b**) Ni@CeGdO$_2$ and (**c**) Ni@CeSmO$_2$.

SEM images of the spent catalysts are shown in Figure 11. As observed by TGA, the Ni@CeGdO$_2$ and Ni@CeSmO$_2$ present a large formation of carbon filaments. Ni@CeO$_2$ catalyst also exhibits the presence of carbon filaments, but in a lower amount compared to Gd and Sm-doped catalysts. The SEM image of Ni@CeZrO$_2$ catalyst does not show carbon filaments, indicating that Zr doping suppresses the carbon deposition.

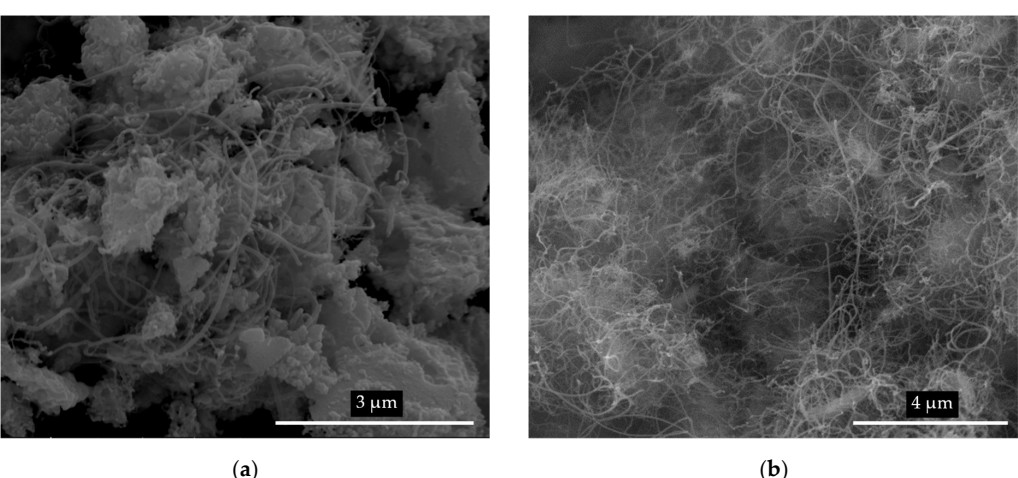

(a)

(b)

**Figure 11.** *Cont.*

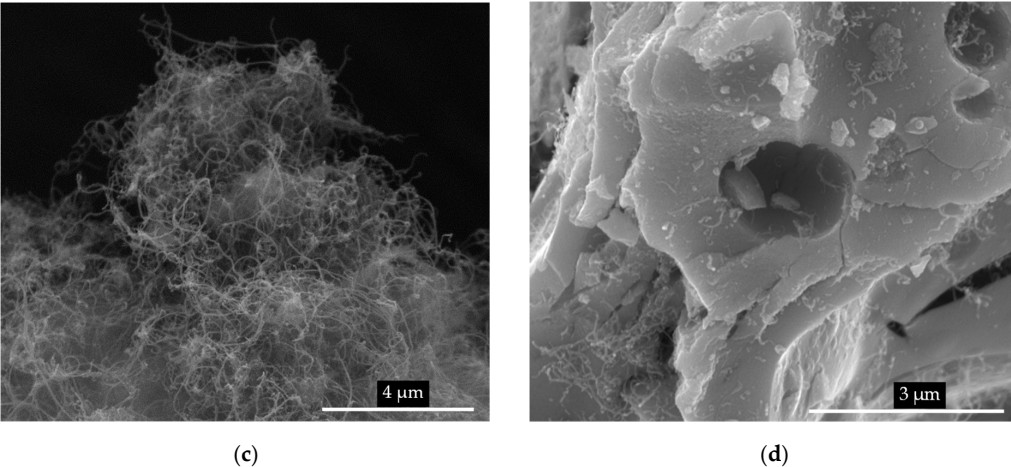

(**c**)　　　　　　　　　　　　　　　　(**d**)

**Figure 11.** SEM images of spent catalysts after DRM at 800 °C for 24 h of TOS: (**a**) Ni@CeO$_2$, (**b**) Ni@CeGdO$_2$, (**c**) Ni@CeSmO$_2$ and (**d**) Ni@CeZrO$_2$.

## 3. Discussion

The mechanism of carbon formation during methane reforming reactions has been extensively studied in the literature [62–67]. According to these studies, the size of nickel particles plays an important role in the nucleation and growth of carbon filaments [48,68–72]. It has been reported that the formation of carbon is a structure sensitive reaction. The solubility of carbon through the metallic Ni particle is determined by its size; the larger the particle, the higher the driven force that promotes carbon diffusion. Recently, we investigated the relationship between the Ni particle size and the type and amount of carbon formed over Ni/CeO$_2$ catalysts in the DRM at 1073 K [48]. The results revealed that the Ni crystallite size affects the type and the rate of carbon formation during DRM. A maximum rate of carbon formation is achieved at around 20–30 nm. Below 10 nm and above 100 nm, the formation of carbon is negligible. Therefore, one of the strategies to suppress carbon formation during DRM reactions is to control the size of the metallic particle using different catalysts preparation methods. The support-embedded Ni approach to control Ni sintering is a promising alternative to design coke-resistant catalysts [9,11]. Such an approach avoids the addition of a second metallic site which can sometimes limit the Ni sintering [73,74]. Our previous work [6] demonstrated that controlling the Ni particle size by embedded process into CeO$_2$ support could decrease the rate of carbon formation during DRM reaction from 9.7 to 1.6 mgC·g$_{cat}$·h$^{-1}$.

In this work, Ni@CeO$_2$, Ni@CeGdO$_2$ and Ni@CeSmO$_2$ catalysts present Ni crystallite size between 9.5–11 nm, which favors the formation of carbon [75]. Ni@CeZrO$_2$ has Ni crystallite (5.7 nm) smaller than the critical size (below 10 nm), which inhibits carbon formation. The doping with Zr enhances CeO$_2$ thermal stability, which contributes to inhibit the growth of the Ni crystallite size during reduction in comparison to the other doped-ceria catalysts.

In spite of the importance of controlling the Ni particle to improve catalyst stability, several authors have reported the formation of carbon on Ni-based core-shell catalysts with high resistance to Ni sintering [12,76–79]. The presence of carbon species may be attributed to the absence of reducible support in the catalyst composition.

The support plays a key role in the CO$_2$ activation and carbon removal mechanism [23,24,80]. In the mechanism for DRM, the dissociation of methane on Ni surface produces hydrogen and highly reactive carbon species (C$_\alpha$), which may follow two reaction pathways: (i) this carbon species reacts with oxygen from the support and the metal surface remains free of carbon deposits; or (ii) it polymerizes to less active carbon (C$_\beta$), then accumulates on the surface as amorphous carbon or dissolves into the Ni lattice, leading to the growth of carbon filaments. Supports with high oxygen mobility promote the oxidation of C$_\alpha$ species before their polymerization, forming CO, avoiding the accumulation of carbon

deposits. The release of oxygen from the support generates oxygen vacancies, in which $CO_2$ is preferentially dissociatively adsorbed as CO and O species, replenishing the oxygen vacancies of the support. Then, the support should have a high oxygen mobility to promote the removal of carbon formed during DRM conditions.

Ceria and ceria-mixed oxides have been largely used as a support to improve catalyst resistance to carbon formation in reforming reactions, especially DRM reaction [33,60,81]. Laosiripojana and Assabumrungrat [81] associated the high resistance to carbon formation observed for Ni supported on $CeO_2$ with high surface area to the higher oxygen mobility of this support, compared to $Ni/Al_2O_3$. For the DRM reaction, the reaction rate between the lattice oxygen from ceria with carbon deposits is improved and the carbon removal mechanism occurs successfully. In this case, the creation of oxygen vacancies is fundamental to promote the $CO_2$ dissociation and the oxygen mobility. Oxygen vacancies can be generated by the ceria reduction process or by the addition of dopants to the ceria lattice.

The addition of Gd, Sm, and Zr may create oxygen vacancies in the ceria structure, but their nature is different. $Gd^{3+}$ and $Sm^{3+}$ are trivalent cations and the partial substitution of $Ce^{4+}$ by them creates oxygen vacancies by charge compensation [28]. The structure is not significantly affected because $Gd^{3+}$, $Sm^{3+}$, and $Ce^{4+}$ have similar ionic radius ($Gd^{3+}$— 1.16 Å; $Sm^{3+}$—1.08 Å; $Ce^{4+}$—0.97 Å). Different behavior occurs when $Zr^{4+}$ is inserted, for which the charge is the same, but the ionic radii is smaller compared to $Ce^{4+}$ ($Zr^{4+}$— 0.84 Å). Therefore, the partial substitution of $Zr^{4+}$ for $Ce^{4+}$ creates oxygen vacancies due to the structural relaxation in the ceria structure [51,82]. Zr has a preference for six-fold coordination, in contrast to the eight-fold coordination observed in the fluorite structure [51]. The change in the structure directly affects the redox properties of ceria.

As observed in the in-situ XRD, TPR, and IOIE experiments, the addition of Zr allows the participation of oxygen atoms from bulk phase in the redox process, increasing the ceria reducibility over the material. The contraction in the lattice creates a driven force of oxygen bulk species to surface [37,51], increasing the number of oxygen species available to be released, which promotes the carbon removal mechanism in the DRM reaction. Oxygen isotopic exchange measurements show that a longer time is required for the exchange process, indicating the migration of oxygen bulk species to the surface [6,7].

In our work, the doping with Gd and Sm creates oxygen vacancies in the surface of the catalysts by charge compensation, quickly reaching the thermodynamic equilibrium for oxygen exchange. Moreover, the replacement of $Ce^{4+}$ by non-reducible ions decreases the amount of reducible material on the catalyst. Therefore, the ceria reduction is not favored, resulting in a low ceria reducibility, as observed by TPR for $Ni@CeGdO_2$ and $Ni@CeSmO_2$ catalysts. The IOIE experiments show that the oxygen vacancies created on these materials are presented more pronounced over the surface, depressing bulk reduction.

Hennings and Reimert [49] evaluated the effect of Gd doped ceria with different concentrations and they observed that the addition of Gd into ceria structure could increase the dynamic oxygen exchange, but it suppresses the OSC value of the catalyst. According to the authors, the formation of vacancies by charge compensation hinders the formation of new vacancies by ceria reduction due to the presence of $Gd^{3+}$ ions and lower $Ce^{4+}$ conversion, leading to low ceria reducibility. Huang et al. [83] evaluated the influence of ceria doping with Sm (SDC) and Gd (GDC) in the DRM reaction. They observed that the GDC catalyst presented a higher density of the surface oxygen vacancies but lower oxygen-ion conductivity, which resulted in lower decoking activity compared to SDC catalyst. The authors attributed the negative effect on oxygen-ion conductivity to the stronger interaction between the oxygen vacancies and the surface O species.

Therefore, in our work, the doping with Gd and Sm reduces the redox cycle, since they have a combination of fewer oxygen atoms and lower ceria reducibility, resulting in a catalyst with high carbon formation compared to the undoped $Ni@CeO_2$. The $Ni@CeZrO_2$ catalyst presents higher ceria reducibility, as observed by TPR, leading to a high amount of bulk oxygen vacancies during the DRM reaction. The creation of bulk oxygen vacancies during the ceria reduction of $Ni@CeZrO_2$ promotes the oxygen-transport and, consequently,

the carbon gasification. Moreover, the CeZrO$_2$ solid solution stabilizes the Ni particle size at DRM condition, below the critical size of 10 nm.

## 4. Materials and Methods

### *4.1. Catalyst Preparation*

The synthesis procedure is described in a previous work [6]. The solution with Ce(NO$_3$)$_3$ and the dopant precursor was prepared with a concentration of metals equal to 1.7 mol·L$^{-1}$ and Ce/dopant molar ratio of 4.0. The dopant precursors were Gd(NO$_3$)$_3$, Sm(NO$_3$)$_3$, and ZrO(NO$_3$)$_2$. To this solution, the appropriate amount of Ni(NO$_3$)$_2$ was added to obtain 10 wt% of Ni. Citric acid solution (6.7 mol·L$^{-1}$) was prepared in another beaker, with a citric acid/metals molar ratio equal to 1.0. Both solutions were mixed and maintained under stirring for 2 h at room temperature. After that, the sol-gel solution was heated up to 70 °C under vacuum (60 mmHg) to remove water and obtain the gel. The material was dried overnight at 100 °C and calcined in two steps: 300 °C for 2 h and 400 °C for 4 h, with a heating rate of 1 °C/min. The resulting catalysts were denominated as Ni@CeO$_2$, Ni@CeGdO$_2$, Ni@CeSmO$_2$, and Ni@CeZrO$_2$.

### *4.2. Characterization*

#### 4.2.1. X-ray Fluorescence

The composition of each catalyst was determined by X-ray fluorescence (XRF), using the RIGAKU RIX-3100 spectrometer. The samples were previously prepared as pellets.

#### 4.2.2. N$_2$ Adsorption

The N$_2$ adsorption was used to measure the BET surface area of the reduced samples. The samples were reduced ex situ at 800 °C for 1 h under pure H$_2$ and passivated for 1 h under 5% O$_2$/N$_2$ at −70 °C. The N$_2$ adsorption was performed in an ASAP 2020 apparatus at −196 °C. The samples were previously degassed at 300 °C under vacuum.

#### 4.2.3. In Situ X-ray Diffraction

The reduction of the samples was followed by in situ X-ray diffraction measurements (in situ XRD) in a Bruker D8 Advance X-ray Powder diffractometer. The apparatus was operated at 40 kV and 40 mA, using CoK$_\alpha$ radiation (λ = 1.790307 Å), equipped with kβ filter (Ni), a positive sensitive detector (VANTEC-1), and worked in scanning mode. All the data were corrected to Cu wavelength (λ = 1.5406 Å) using Bragg's law for comparison with the data from the literature. The samples were reduced under a 10 % H$_2$/He mixture (30 mL/min) from room temperature to 800 °C at 10 °C/min. The diffractograms were collected in the 2θ range of 10–80°, using a scan rate of 0.05°/step and a scan time of 2 s/step at different temperatures: room temperature; 300, 500, and 800 °C.

#### 4.2.4. Raman Spectroscopy

The RAMAN spectra were recorded at room temperature using a Horiba LabRam HR-UV800/Jobin-Yvon spectrometer equipped with a 532 nm wavelength laser.

#### 4.2.5. Temperature-Programmed Reduction

Temperature-programmed reduction (TPR) was performed in an apparatus equipped with a thermal conductivity detector. The sample (300 mg) was previously oxidized at 400 °C for 1 h under air (30 mL/min). Then, the sample was cooled down to room temperature and heated under 10% H$_2$/Ar mixture (30 mL/min) from room temperature up to 1000 °C (10 °C/min).

#### 4.2.6. $^{18}$O$_2$/$^{16}$O$_2$ Isotopic Exchange

The experiments were carried out in a closed recycling system connected to a Pfeiffer Vacuum quadrupole mass spectrometer and a vacuum pumper. Hence, 20 mg of catalyst was pre-treated under $^{16}$O$_2$ flow (50 mL/min, 500 °C, 1 h) and evacuated for 1 h.

For the isothermal oxygen isotopic exchange (IOIE), the reaction occurred at 400 °C. It was inserted in the reactor 55 mbar of pure $^{18}O_2$ ($\geq$99 at.%, ISOTEC) and each iso-topomer concentration was analyzed by monitoring the following m/z signals: 32 ($^{16}O_2$), 34 ($^{18}O^{16}O$), 36 ($^{18}O_2$).

The following equations were used to calculate the amount of oxygen exchanged ($N_e$) and the atomic fraction of $^{18}O_2$ in the gas phase at the time t ($\alpha_g^t$). $N_g$ is the total number of oxygen atoms in the gas phase, w is the weight of catalyst, and $\alpha_g^0$ is the initial atomic fraction of $^{18}O_2$ in the gas phase. Expressions for the calculation of $\alpha_g^t$ and $N_e$ are given by Equations (1) and (2) respectively.

$$\alpha_g^t = \frac{\frac{1}{2}P_{34}^t + P_{36}^t}{P_{36}^t + P_{34}^t + P_{36}^t} \tag{1}$$

$$N_e = \left(\alpha_g^0 - \alpha_g^t\right) N_g \frac{1}{w} \tag{2}$$

*4.3. Catalytic Test*

The DRM reaction was carried out in a fixed-bed quartz reactor at atmospheric pressure. Prior to the reaction, the samples (20 mg of catalyst diluted with SiC: SiC/catalyst = 1.5) were reduced in situ under $H_2$ (30 mL/min) from room temperature to 800 °C (10 °C/min) for 1 h and purged with $N_2$ (30 mL/min) for 30 min. After this treatment, the reactant mixture with $CH_4/CO_2$ molar ratio of 1.0 at a flow rate of 100 mL/min was flowed through the catalyst for 24 h at 800 °C.

The reaction products were analyzed by gas chromatography (Agilent 6890) equipped with a thermal conductivity detector and Carboxen 1010 column (Supelco). The equations for $CH_4$ conversion (Equation (3)), $CO_2$ conversion (Equation (4)), and $H_2/CO$ molar ratio (Equation (5)) are described below:

$$X_{CH_4} = \frac{\left(F_{CH_4}\right)_{in} - \left(F_{CH_4}\right)_{out}}{\left(F_{CH_4}\right)_{in}} \times 100 \tag{3}$$

$$X_{CO_2} = \frac{\left(F_{CO_2}\right)_{in} - \left(F_{CO_2}\right)_{out}}{\left(F_{CO_2}\right)_{in}} \times 100 \tag{4}$$

$$\frac{H_2}{CO} = \frac{\left(F_{H_2}\right)_{out}}{\left(F_{CO}\right)_{out}} \tag{5}$$

where $F_{CH_4}$, $F_{CO_2}$, $F_{H_2}$, and $F_{CO}$ represent the molar flowrates of $CH_4$, $CO_2$, $H_2$, and CO, respectively.

*4.4. Carbon Analysis*

4.4.1. Scanning Electron Microscopy

The post-reaction samples were analyzed using a field emission scanning electron microscope (SEM) (Quanta FEG 450 FEI). The microscope was operated with an accelerating voltage of 20 kV.

4.4.2. Thermogravimetric Analysis

The quantification of carbon deposits in the spent samples was performed by thermogravimetric analysis (TGA). The experiment was performed in a TA Instrument apparatus (SDT Q600) using 10 mg of sample. The sample was heated from 25 to 1000 °C at a heating rate of 10 °C/min under synthetic air (100 mL/min) while monitoring the weight variation.

## 5. Conclusions

The sol-gel method employed to synthesize Ni@CeO$_2$, Ni@CeGdO$_2$, Ni@CeSmO$_2$, and Ni@CeZrO$_2$ catalysts led to the formation of Ni nanoparticles embedded into ceria-doped oxide. The thermal stability of Zr avoids Ni metal sintering at high temperature (800 °C). Although the addition of dopants creates oxygen vacancies in the material, the ceria reducibility differs depending on the dopant used. The lattice distortions caused by Zr insertion in the ceria lattice enhance the bulk oxygen diffusion towards the surface, increasing ceria reducibility. The doping with Gd and Sm only creates oxygen vacancies by charge compensation, saturating the surface with oxygen vacancies and decreasing ceria reducibility in comparison to pure ceria and Zr-doped ceria. As a consequence, high carbon formation is detected for Sm and Gd-doped catalysts. Therefore, the combination of (i) small Ni crystallite size and (ii) high ceria reducibility promotes the balance between CH$_4$ decomposition and carbon gasification, resulting in the suppression of carbon deposits under DRM reaction and high activity for Ni@CeZrO$_2$ catalyst.

**Author Contributions:** Conceptualization, F.S.T., F.B.N. and N.B.; investigation, A.L.A.M.; writing—original draft preparation, A.L.A.M.; writing—review and editing, A.L.A.M., R.C.R.-N., F.E., F.S.T., F.B.N. and N.B.; supervision, F.S.T., F.B.N., F.E. and N.B.; project administration, F.B.N. and N.B.; funding acquisition, F.B.N. and N.B. All authors have read and agreed to the published version of the manuscript.

**Funding:** This research was funded by Coordenação de Aperfeiçoamento de Pessoal de Nível Superior (CAPES—Finance code 001; 88881.142911/2017-01), the Conselho Nacional de Desenvolvimento Científico e Tecnológico (CNPq—303667/2018-4; 305046/2015-2; 302469/2020-6; 310116/2019-82; CNPq-SisNANO—442604/2019-0), the Fundação de Amparo à Pesquisa do Estado do Rio de Janeiro (FAPERJ—26/010.253/2016; E-26/202.783/2017) and the Comité Français d'Évaluation de la Coopération Universitaire et Scientifique avec le Brésil (COFECUB—Ph-C 912/18) for scholarship and financial support.

**Acknowledgments:** Sandrine Arii and Nadia Guignard are gratefully acknowledged for in situ XRD and RAMAN experiments respectively. European Union (ERDF) and Région Nouvelle Aquitaine region are also gratefully acknowledged for their financial support. F.B.N. also thanks the French government through the Programme Investissement d'Avenir (I-SITE ULNE/ANR-16-IDEX-0004 ULNE) managed by the Agence Nationale de la Recherche, LIA CNRS France-Brazil "Energy & Environment", Métropole Européen de Lille (MEL) and Region Hauts-de-France for "CatBioInnov".

**Conflicts of Interest:** The authors declare no conflict of interest.

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
