# Peer review of "Effect of Metal Dopant on the Performance of Ni@CeMeO2 Embedded Catalysts (Me = Gd, Sm and Zr) for Dry Reforming of Methane"

_methane, doi:10.3390/methane1040023_

Round 1

Reviewer 1 Report

The manuscript fits well with the scope of the journal and the results are well presented. I suggest only small improvements.

For the DRM test, the Thermodynamic has to be presented in order to show the potential of such materials

also I suggest to the authors to comment a bit more the following part

"The CO2 conver-350 sion is higher than CH4 conversion for all catalysts while the H2/CO molar ratio is lower 351
010203040506070809001020304050 Ne (1020 at.g-1)Time (min) Ni@CeO2 Ni@CeGdO2 Ni@CeSmO2 Ni@CeZrO2
Methane 2022, 1, FOR PEER REVIEW 11
than 1.0. These results suggest the occurrence of the reverse water-gas shift (RWGS) reac- 352
tion, which is thermodynamically favored at the reaction conditions used. Similar result 353
was reported by other authors [30,40,53]."

Finally, it would be good, also in the catalytic part to show the impact of each promoting element in the CeOx support and link it with catalytic activity

Reviewer 2 Report

In this manuscript, the authors investigated the influence of metal dopant (Gd, Sm and Zr) into CeO2 structure for the dry reforming of methane over Ni nanoparticles embedded catalysts. They showed that the doping with Zr improved thermal stability of the catalyst, leading to the formation of small Ni nanoparticles, while Ni metal sintering was observed for Ni@CeO2, Ni@CeGdO2 and Ni@SmO2, according to in situ XRD under reduction conditions. Furthermore, the Ni@CeZrO2 catalyst did not present coke formation because of smaller Ni crystallite size and higher ceria reducibility. I think that is a very interesting experimental study, however, some issues should be improved. I recommend a publication of this work to Methane journal after the authors consider the following major revisions.

Comment #1

The authors need to improve on their bibliography. I think that they should add some references in order to enrich the introduction section. In particular, during reforming reaction, different types of carbon are developing on the catalyst surface affecting the catalytic activity and stability. The authors could add the most important findings of the following research studies.

The authors could add the most important findings of the following research studies.

[1].         Yuxia Ma, Yuyao Ma, Jiajie Li, Zhengmao Ye, Xun Hu, Dehua Dong, Electrospun nanofibrous Ni/LaAlO3 catalysts for syngas production by high temperature methane partial oxidation. International Journal of Hydrogen Energy 47 (2022) 3867-3875

[2].         N.D. Charisiou, G. Siakavelas, L. Tzounis, V. Sebastian, A. Monzon, M.A. Baker, S.J. Hinder, K. Polychronopoulou, I.V. Yentekakis, M.A. Goula, An in depth investigation of deactivation through carbon formation during the biogas dry reforming reaction for Ni supported on modified with CeO2 and La2O3 zirconia catalysts. International Journal of Hydrogen Energy 43 (2018) 18955-18976

Comment #2

The authors need to be clearer on the motivation behind their work. Which is the innovation and what are the new aspects being introduced on this research topic?

Comment #3

What characterization technique did the authors used for the measurement of Ni and promoters’ content?

Comment #3

Is it possible for the author to investigate the oxygen vacancies via Raman spectroscopy?

Comment #5

Is it possible for the authors to calculate the population of the basic sites via TPD-CO2 experiments? Do they play a role in the activity? Explanation is necessary.

Comment #6

As can be observed in Figure 6, Ni/CeZrO2 presents the lower catalytic activity compared to the other catalytic systems. Furthermore, the authors reported that Ni/CeZrO2 presents the lower carbon deposition. I think that the catalytic deactivation is due to the metal particle sintering. The authors should investigate the possibility of the agglomeration of metallic nickel via XRD on TEM analysis on the spent samples.
